# Impact of Drug Pressure versus Limited Access to Drug in Malaria Control: The Dilemma

**DOI:** 10.3390/medicines9010002

**Published:** 2022-01-04

**Authors:** Chinedu Ogbonnia Egwu, Nwogo Ajuka Obasi, Chinyere Aloke, Joseph Nwafor, Ioannis Tsamesidis, Jennifer Chukwu, Sunday Elom

**Affiliations:** 1PharmaDev, UMR 152, Université de Toulouse, IRD, UPS, 31400 Toulouse, France; 2Medical Biochemistry, College of Medicine, Alex-Ekwueme Federal University, Ndufu-Alike Ikwo, P.M.B. 1010, Abakaliki 482131, Nigeria; naobasi@yahoo.com (N.A.O.); alokec2002@yahoo.com (C.A.); elomso@yahoo.com (S.E.); 3Protein Structure-Function and Research Unit, School of Molecular and Cell Biology, Faculty of Science, University of the Witwatersrand, Braamfontein, Johannesburg 2050, South Africa; 4Anatomy, College of Medicine, Alex-Ekwueme Federal University, Ndufu-Alike Ikwo, P.M.B. 1010, Abakaliki 482131, Nigeria; jnwafor49@gmail.com; 5Department of Prosthodontics, School of Dentistry, Faculty of Health Sciences, Aristotle University of Thessaloniki, 54124 Thessaloniki, Greece; johntsame@gmail.com; 6John Hopkins Program on International Education in Gynaecology and Obstetrics, Abuja 900281, Nigeria; gennytimah@gmail.com

**Keywords:** malaria, drug-pressure, limited-access, antimalarials, treatment

## Abstract

Malaria burden has severe impact on the world. Several arsenals, including the use of antimalarials, are in place to curb the malaria burden. However, the application of these antimalarials has two extremes, limited access to drug and drug pressure, which may have similar impact on malaria control, leading to treatment failure through divergent mechanisms. Limited access to drugs ensures that patients do not get the right doses of the antimalarials in order to have an effective plasma concentration to kill the malaria parasites, which leads to treatment failure and overall reduction in malaria control via increased transmission rate. On the other hand, drug pressure can lead to the selection of drug resistance phenotypes in a subpopulation of the malaria parasites as they mutate in order to adapt. This also leads to a reduction in malaria control. Addressing these extremes in antimalarial application can be essential in maintaining the relevance of the conventional antimalarials in winning the war against malaria.

## 1. Introduction

The control of malaria has stalled in the last five years [1,2]. There is need for a continuous review of the control strategies. Antimalarials remain one of the formidable tools in malaria control. Both reduced access to antimalarials by the populace and pressure from these drugs on malaria parasites are important factors that play different roles in the overall malaria control. Limited access to drug and drug pressure are two sides of a coin that can individually and/or collectively lead to treatment failure. While reduced access to drugs negatively impact malaria control via increased spread/transmission, drug pressure facilitates the generation of resistant *Plasmodium* which results in treatment failures, respectively. Some antimalarials such as chloroquine (CQ) have lost their place in malaria control due to resistance development, and this fate befalls several other antimalarials [3,4]. Treatment failure can result from drug resistance (often from drug pressure), high baseline parasite density, childhood (age less than five), incorrect dosing, non-compliance with duration of dosing regimen, poor drug quality and drug interactions leading to decreased drug exposure [5]. Other factors that can lead to reduced access may include: cost of the drug, reduced or poor access to health facilities, lack of free drug or functional health insurance scheme, and some socio-cultural as well as religious characteristics of a people (e.g., religious beliefs and traditions).

This review seeks to assess the impact of these contrasting factors, limited access to drug and drug pressure, on the overall malaria treatment failure vis-à-vis its control.

## 2. Overview of the Current Antimalarial Portfolio

Before the advent of refined antimalarials, the use of herbs had been the hallmark of treatment, as exemplified by the *Cinchona* bark from which quinine was extracted and identified in 1820 by the French chemists, J. Pelletier and J. Caventou [6]. Today, several drugs are currently used for the treatment of malaria. These drugs, for convenience, are classified into four major classes: antifolates, quinolines (aryl-amino alcohols, 4-aminoquinolines, bis-quinoline and naphthyridine), hydroxynaphtoquinone and endoperoxides (Table 1). The antifolates interfere with folic acid synthesis which is essential in nucleic acid metabolism [7]. The commonest antifolates in use are pyrimethamine and sulfadoxine. The main mode of action of quinolines and endoperoxides is targeting of the heme detoxification process put in place during the digestion of hemoglobin by the parasite [8,9]. The quinolines are mainly represented by chloroquine, mefloquine, amodiaquine, lumefantrine and quinine [10]. For the endoperoxides, artemisinin and its derivatives (artemether, artesunate, arteether) are the main examples. The endoperoxides generally increase the oxidative stress level, attacking essential biomolecules such as proteins, nucleic acids and lipids in the parasite, which leads to the death of the parasites [11]. The hydroxynaphtoquinones inhibit the respiratory chain of the parasite [12]. The commonest hydroxynaphtoquinone in use is atovaquone.

In order to avoid the emergence of parasite resistance, drug combinations are strongly advocated for malaria treatment. The recommended combination is artemisinin-based therapy (ACT) (Table 2), although other combinations such as atovaquone/proguanil and sulfadoxine/pyrimethamine are also available. For example, to protect endoperoxide activity and their partner drugs against parasite resistance, since 2001 the WHO has recommended the use of artemisinin derivatives with one or two partner drugs, which are chosen according to the differences in their pharmacokinetic properties. As a result of the failure that has been reported with dihydroartemisinin-piperaquine (DHA-PPQ) and other ACTs in Southeast Asia (Table 2), the use of triple combination therapy has been put forward as a more reliable approach going by the efficacy outcome obtained from atovaquone-proguanil–artesunate [13]. However, the cost of the triple combination, atovaquone–proguanil–artesunate, may limit its application in low-to-medium income countries were malaria is endemic.

Prevention is said to be better than cure. Some antimalarials can be used for chemoprevention. This reduces the risk of developing overt malaria signs and symptoms. This also reduces the mortality risk associated with malaria especially in children under five years, pregnant women and travelers, who do not live in malaria endemic countries. Atovaquone/proguanil and sulfadoxine/pyrimethamine are among the antimalarials that can be used in chemoprevention of malaria (Table 1) [14,15].

## 3. Limited Access to Drugs

For effective treatment of diseases, access to the right medications is a prerequisite, among several other factors. The availability of quality and affordable drugs in sufficient quantity is an essential element in universal health coverage. Several health agencies including the World Health Organization (WHO) have since their establishment worked to increase access to medications. Unfortunately, these efforts have been frustrated by a number of factors, ranging from economic to sheer wickedness on the part of businessmen. Lack of access to medicines causes a cascade of misery and suffering from no relief for a particular disease. Several factors are contributory to the limited access to drugs. Limited access to medicines caused by other factors promotes the sale of counterfeit drugs with sub-clinical quantities and the use of monotherapy. More so, an intense and extensive migrant labor system worsens access to drugs [34]. These factors are discussed below (Figure 1).

### 3.1. Counterfeiting/Drug Quality

The quality of drug refers to the suitability of either a drug substance or drug product for its intended use as described during an FDA international conference on Harmonization [35]. The quality of a drug is anchored on three features: identity, strength and purity for the intended use [36]. The quality of a drug must be assured and maintained until it gets to the end users (patients). Regrettably, there is a great deal of interplay between drug manufacture and delivery to the end users which compromises the drug quality. Such interplay includes but not limited to: compromised drug manufacture (sub-therapeutic active ingredients, non-compatible excipients and lack of good manufacturing practices (GMP)). Inaccessibility and high price of quality ACTs, poor transportation and harsh storage conditions, wrong dispensing (mistakes by dispensers) as well as lack/or inadequate regulation from regulatory agencies, contribute immensely to counterfeiting of antimalarials. The use of poor quality antimalarials can cause a drop in efficacy and reduce parasite clearance, which are characteristics of drug resistance and eventual drug failure [37]. Drugs against infectious diseases such as malaria are among the most falsified drugs in the market especially in developing countries, which fall under malaria endemic regions. According to a WHO account, one in every ten drugs in developing countries are either substandard or falsified [38]. Quality tests conducted on some artemisinin derivatives in Africa failed the tests [39]. This calls for more regulation from the government of these affected countries to ensure strict compliance to GMP.

Even when the drugs meet the required standard after manufacture, the means of transportation to the pharmacies or hospitals are often inadequate. This is mostly the case with drugs that are unstable under certain temperature ranges. The impact of poor quality drugs is huge; about 122,000 deaths were estimated in children under five in sub-Sahara Africa and therefore should be taken seriously [40]. The counterfeiting of antimalarials limits the availability of the required amount of drugs for effective treatment and hence can culminate in treatment failure and reduction in malaria control.

### 3.2. Cost of Drug

Most, if not all, the novel antimalarials hit the market at exorbitant prices as premium brands. This adversely affects the affordability by the deserving masses, thereby limiting access to the drugs. Some patients may attempt to buy some doses. Consequently, there will be an incomplete or non-treatment of malaria across different categories of patients. Incomplete drug regimens can lead to treatment failure on subsequent administration of complete doses. Sometimes, the lack of resources from the patients can result in the use of less effective antimalarials.

Both non-treatment and incomplete treatment continually make the subjects harbors for the malaria parasites, which enhances transmission to other people, worsening malaria control. The WHO considers health insurance as a veritable tool in ameliorating the effect of the cost of essential drugs [41,42]. Ordinarily in developed countries, the cost burden of drugs are borne by health insurance; however, in most of the low-to-medium income countries (LMIC) where malaria is endemic, health insurance utilization or its awareness is still relatively low [43]. Most patients still pay for their drugs out of the pocket. Attention must be paid to the cost of drugs as treatment outcome or efficiency is tightly linked to the cost of prescription. The high cost of some antimalarials, especially ACTs, raises the risk of utilizing counterfeit or sub-standard brands due to switches at the pharmacy because of their cheapness. This can lead to treatment failure which may be progressive, affecting the regimen even when the right drugs are eventually used.

To ameliorate the impact of high cost of novel antimalarials, most malaria endemic countries benefit from antimalarial donations which are made available to malaria subjects free of charge. Increasing access to antimalarials through reduced cost can improve the treatment outcome and reduce transmission significantly. This will reduce overall malaria burden.

### 3.3. Drug Storage/Transportation

Antimalarials remain viable at certain temperatures. However, this is not the case in the course of movement of these drugs to some rural communities. The drug gets to the end users in a degraded form with reduced efficacy. Improved means of transportation of drugs and storage can therefore retain the quality of drugs as they get to the end users. The right environmental controls such as temperature, light, and humidity, conditions of sanitation, ventilation, and segregation, which affect the drug stability and quality [44], must be regulated during drug transit and storage.

### 3.4. Drug Interactions

The drugs that reach the systemic circulation can be grossly affected by their interaction with substances ingested alongside the drugs. Such interactions affect the absorption, distribution, metabolism, and/or excretion of drugs, which can reduce the clinical efficacy of the drug. The interaction could be additive, synergistic or antagonistic. Such substances are mainly either food (drug-food interaction) or other drugs (drug-drug interaction). Synergistic interactions can improve the efficacy of the index drug. Antagonistic interactions can reduce the drugs available at the site of action, in other words, reducing access to the drug. For example, some antiviral and antibacterial agents administered concurrently with antimalarials result in antimalarial failure [45]. While some interactions decrease the availability and activity of some antimalarials, others may increase their activity [46,47]. For instance, imatinib or other Syk kinase inhibitors which potentiate artemisinin combination therapies [48]. Due to the interaction of antimalarials with other drugs or food, patients must seek proper medical counsel before taking any antimalarial with other drugs. The interaction of antimalarials and other substances is shown in Table 3.

### 3.5. Use of Antimalarials as Monotherapies

Different antimalarials have different pharmacokinetic properties including short half-lives. The recommended combination is artemisinin-based therapy (ACT) (Table 2), although other combinations such as atovaquone/proguanil and sulfadoxine/pyrimethamine are also available. Atovaquone/proguanil and sulfadoxine/pyrimathamine are used mainly in prophylaxis (malaria chemoprevention) in travelers and pregnant women, respectively [14,15]. The combination makes up for the shortcomings (such as short half-life of artemisinin) of each drug in the combination, ensuring effective exposure time [49]. The currently recommended six ACTs from WHO are: artemether-lumefantrine (AL), dihydroartemisinin-piperaquine (DHA-PPQ), artesunate-amodiaquine (AS-AQ), artesunate-mefloquine (AS-MQ), artesunate-sulfadoxine-pyrimethamine (AS-SP) and artesunate-pyronaridine (AS-PY) [32]. Regrettably, in some places and for some reasons, people still use these antimalarials as monotherapies. The exposure of the parasite to only one drug makes it easier for the parasite to develop resistance to such antimalarial drugs, leading to a reduction in malaria control. The effective use of these ACTs is therefore strongly advocated to improve access to drugs.

### 3.6. Hoarding of Drugs by Corrupt Officials

There are several malaria programs intended to increase access to drugs by different governmental and non-governmental agencies. These agencies either subsidize the drugs or make them entirely free [2]. These drugs go through some middlemen before getting to the intended targets. Regrettably, sometimes the drugs get stockpiled in the hands of these middlemen, who do so for several reasons ranging from profit-making to sheer wickedness. They hoard the drug in order to resell them to make gains. This leaves the poor masses without the usually subsidized or free drugs.

## 4. Drug Pressure

### 4.1. Defining the Phenomenon of Drug Pressure

The malaria parasites are often exposed to different antimalarials to eliminate them and prevent their spread. However, continuous exposure to these drugs (drug pressure) can lead to a reduced sensitivity to the same drugs in subsequent administrations. The mechanism of developing reduced sensitivity may differ for each molecule (and sometimes overlap) but are commonly due to a mutation, which could range from a point to multiple mutations [18,26,27,28]. The drug does not cause the mutation but leads to a selection of the mutant population. The mutant population can transfer the resistance phenotype to subsequent generations of the parasite. The mutations cause the evolution of a subpopulation of the parasite that can survive the drug administration. However, some of the population of the parasites may remain susceptible to the drugs. Several schools of thought had opined that massive administration of antimalarials would lead to the elimination of malaria. Contrastingly, this exacerbates the situation by leading to resistance development due to drug pressure [50,51].

### 4.2. Genes Affected by Drug Pressure from Antimalarials

#### 4.2.1. Antifolates

Resistance to antifolates is associated with a point mutation in dihydrofolate reductase (*pfdhfr*) and dihydropteroate synthase (*pfdhs*) respectively. Intense use of antifolates leads to the selection of the resistance phenotype, which are transferable to future parasite generations. This phenomenon varies from molecule to molecule based partly on their period of exposure. The longer acting antifolates, such as pyrimethamine and sulfadoxine, lead to the selection of more resistance phenotype than the short acting ones such as chlorproguanil and dapsone [52].

#### 4.2.2. Quinolines

Resistance to quinolines is due to mutations in the gene of the transmembrane protein and multidrug resistance genes, *Plasmodium falciparum* chloroquine resistance transporter (*pfcrt*) and *Plasmodium falciparum* multidrug resistance transporter (*pfmdr*), respectively [18,23]. These mutations result in drug efflux, reducing the availability of the drug in the food vacuole for its pharmacological activity, consequently reduced heme detoxification, which is the ultimate mechanism of action of this class of antimalarial [53].

#### 4.2.3. Hydroxynaphtoquinones

For hydroxynaphtoquinones, mainly represented by atovaquone, resistance is due to a point mutation in the *Plasmodium falciparum* cytochrome b (*pfcytb*), which results in the modification of the binding site on complex III/cytochrome b of the electron transport chain [26,28]. This leaves this complex III in the electron transport chain of the mitochondrion uninhibited; hence, parasite survival.

#### 4.2.4. Endoperoxides

Resistance to the current gold standard drug, artemisinin, was first reported in 2008 and then confirmed in 2009 at the Cambodia-Thailand border after artesunate monotherapy, which was characterized by a prolonged parasite clearance (more than 3 days) [4,54,55]. This spread westward to other parts of Southeast Asia. A recent report of a local emergence of K13 mutation and expansion of Pfkelch13 561H lineage in Rwanda raises concerns over malaria control, even though this has not been linked to any clinical treatment failure [56]. The mechanism of resistance to artemisinin is continuously debated. However, pockets of available evidence show that this happens through a quiescence-based phenomenon linked to mutations in the Pfk13 propeller gene [57]. Mutations in the *Plasmodium falciparum* Pfk13 gene are associated with delayed parasite clearance in response to artemisinin-based treatment. Quiescence is a complex phenomenon where the parasites enter into a dormant/non-dividing state under a drug pressure, only to grow back once the drug is removed (Figure 2). Results from artemisinin-resistant *Plasmodium falciparum* exposed to high artemisinin concentrations showed that artemisinin pressure induces developmental arrest of a subpopulation of very young (0–3 h) ring stages of the parasite, causing them to go into a quiescent state, while killing all other stages. This was demonstrated on both laboratory model (F32-ART) and field isolates of resistant *Plasmodium falciparum* [27,58,59]. Since only the ring stages (other stages spared) of the parasite are affected, the resistance development to artemisinins is said to be a partial resistance.

Several efforts have been put in place and ongoing, in order to explain this phenomenon. During the quiescent state, the metabolism of the parasites are drastically reduced and supported by the remaining mitochondrial activity and the implementation of a lipid metabolism (Type II fatty acid synthesis (FASII)) in the apicoplast [60,61]. The key modifications in metabolism during the quiescent state include but not limited to: reduced hemoglobin endocytosis; acceleration of the unfolded protein response (UPR) pathways; improved ability to manage oxidative damages; changed DNA replication, increased basal level of phosphorylated elf2α and/or increase in levels of phosphatidylinositol-3-phosphoate [62,63,64].

### 4.3. Cross Resistance Due to Drug Pressure

Cross resistance is a phenomenon whereby a parasite strain which is less sensitive to a particular molecule expresses similar reduced responsiveness to other molecules. These molecules may be from the same class or different classes. This is quite common when the molecules have some mechanism of action (MOA) in common, as exemplified by quinolines and endoperoxides which interfere with heme detoxification, in addition to other plausible mechanisms [9,65]. Endoperoxides are activated by heme and the activated endoperoxides can attack several biomolecules including heme, preventing heme polymerization to the non-toxic form, hemozoin. Witkowski et al., demonstrated that exposure of *Plasmodium* to pressures from artemisinin and chloroquine, respectively, leads to the acquisition of resistance with similar phenotypes [66]. These strains become less sensitive to quinolines (chloroquine, mefloquine and quinine) and endoperoxides (artemisinin, artesunate, artemether and artemisinin) [66]. This finding was corroborated by Cui et al. who demonstrated that DHA induced resistant strains also showed reduced susceptibility to lumefantrine, chloroquine, amodiaquine, piperaquine, mefloquine and quinine [67]. The cross resistance to antimalarials jeopardizes malaria control as existing molecules and those in development, which share some mechanism of action in common may be bound to fail in no distant time.

### 4.4. Overcoming Drug Pressure

#### 4.4.1. Proper Diagnosis before Treatment

To prevent misapplication of drugs for wrong indications, proper diagnosis is very essential [68]. There are five diagnostic approaches for malaria, namely presumptive, serological, microscopic, dipstick/rapid diagnostic test (RDT) and PCR-based approaches [5]. In some malaria endemic regions, access to diagnostic tools is limited. In such places, the presumptive approach is still used due to lack of diagnostic facilities in order to commence treatment immediately. The dipstick approach is the fastest and easy-to-use tool. Microscopy remains the gold standard in most clinic for accurate malaria diagnosis. Unlike the dipstick approach, microscopy requires trained experts to execute. Serological approach is not recommended for acute infections due to the delay in the release of antibodies. Serological approach is therefore mainly used for detection of past exposure to malaria parasites. Lastly, The PCR-based approach is a molecular tool mainly used as a confirmatory test and for malaria surveillance purposes. Prompt diagnosis via any of these approaches can ensure a targeted treatment with desired results.

The role of prompt diagnosis cannot be underestimated in malaria control. Prompt diagnosis ensures early detection of the parasites and gives room for immediate treatment of the disease. This reduces the parasite load in the patients, reducing transmission to other persons. This has a huge positive impact in the overall malaria control strategy. An accurate diagnosis must be made as quickly as possible in order to enhance the overall treatment outcome.

#### 4.4.2. Treatment with Only Optimal Doses

To prevent selection of subpopulations that will survive high doses and become non-sensitive to pharmacologically relevant doses, it is pertinent that treatments are initiated with only optimal doses which will not mount pressure on the parasites. This is particularly important as findings have shown that increased pressure due to high doses can lead to the development of resistance [66,67].

#### 4.4.3. Restrictive Drug Use

Since drug pressure is the single most important cause of resistance, restricting drug prescription and usage will go a long way in reducing resistance development and spread [5]. This calls for the application of the tools of malaria control which focus on malaria prevention. Evidence shows that the use of insecticide-treated bed nets is the single most effective tool in malaria control strategies, contributing not less than 68% in the success recorded between 2000–2015 [69] (Figure 3). Increased use of insecticide spray and insecticide-treated bed nets should be massively encouraged to reduce the risk of having full blown malaria that will prompt antimalarial use.

#### 4.4.4. Recycling of Antimalarials

Some strains of *Plasmodium* which were up until now resistant to some antimalarials are becoming sensitive again. A notable example is the regained efficacy of chloroquine against *Plasmodium* in Malawi and other parts of Africa [70,71]. This occurrence gives credence to the notion that the parasites regain sensitivity to the drugs when the drug pressures are removed [57]. This could open door for the recycling of antimalarials as already suggested by Dipanjan et al. [13], consequently reducing the need for the search for newer antimalarials, which is capital intensive and time consuming. A regular surveillance is necessary to detect any reduction in efficacy of antimalarials due to drug pressure for their immediate withdrawal and future recycling. This approach will maintain the tempo in curbing malaria morbidity and mortality. For recycling to be adopted as an answer to incessant resistance to antimalarials, a lot of questions must be addressed. Such questions include but are not limited to: “What is the average time needed by an antimalarial pressure to induce resistance?”, “What are the genetic diversities of the populace that could be considered?” and/or “what is the cost/benefit ratio of withdrawal and recycling of a particular drug?” Addressing these critical questions can make the timing right for recycling of each antimalarial.

#### 4.4.5. Drug Resistance Reversal

Each resistance development is linked to a distortion/mutation in one or more biochemical pathways. The reversal of such distortions could be exploited in continually combating resistance to antimalarials. For instance, CQ resistance is linked to high efflux of CQ from the food vacuole, which prevents CQ from attaining a toxic concentration against the parasite [20]. The blockade of this efflux by verapamil (a calcium channel blocker) and desipramine (a tricyclic antidrepressant), respectively, has been shown to reverse chloroquine resistance [72,73]. In the same vein, a co-incubation of mefloquine and penfluridol increases the susceptibility of mefloquine resistant strains to mefloquine.

Resistance to antimalarials have been shown to cause increase in the antioxidant capacity of the resistant malaria parasites [67,74,75]. For artemisinin and its derivatives, the use of antioxidant inhibitors have shown promising results in reversing resistance to artemisinin [76] through a synergistic means.

## 5. Conclusions

Malaria control is hanging in the balance with the use of available tools of control, especially from the antimalarials. Under- or over-application of antimalarials leads to a similar consequence of reduced malaria control. It is a dilemma (the devil and the deep blue sea) that must be solved for effective malaria control. Adequate drug access via the use of intense regulations to reduce counterfeiting, proper medical advice to avoid unfavorable drug interactions, effective use of acts and reduction in cost of drugs can improve drug access to the malaria patients and prevent the consequences of failures. On the other hand, proper diagnosis before drug use, recycling of drugs, restrictive drug use and use of only optimally relevant doses in treatment can ameliorate failure due to drug pressures. Implementation of the suggestions made in this review may be the panacea to this puzzle in malaria control.

## Figures and Tables

**Figure 1 medicines-09-00002-f001:**
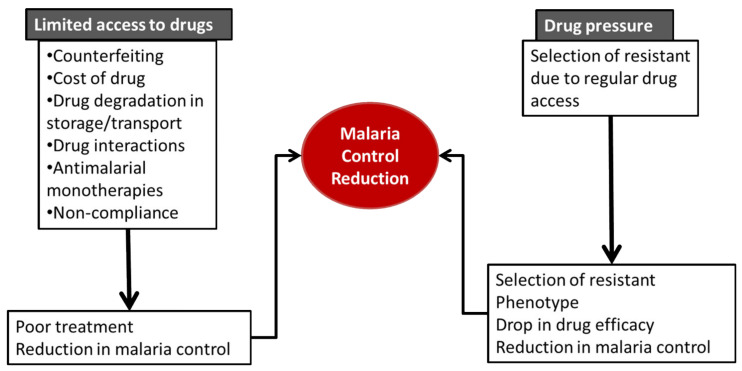
Impact of limited drug access and drug pressure on malaria control.

**Figure 2 medicines-09-00002-f002:**
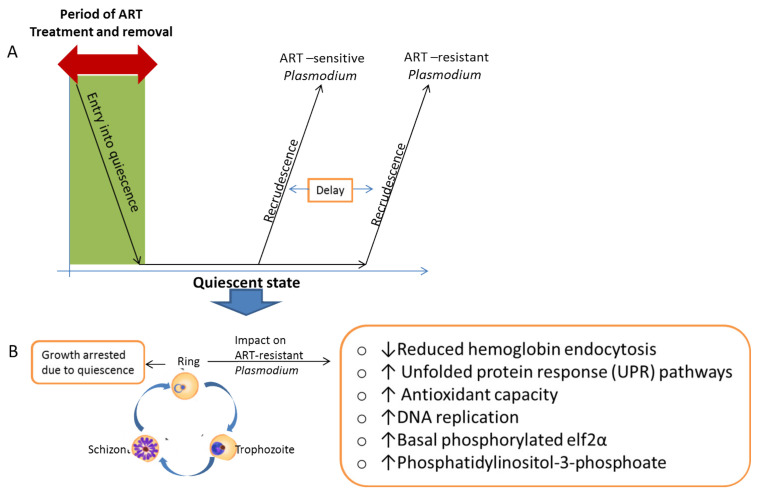
Resistance development selection due to artemisinin drug pressure. The drug pressures lead to the selection of resistant parasite phenotypes conferred by gene mutations that can be passed on to the next generations. (**A**) The impact of ART exposure on ART-sensitive and ART-resistant *Plasmodium*, where a subpopulation goes into quiescence and recrudesces when the ART is removed, where there is a delay in the recrudescence of the resistant strains. (**B**) Characteristics of the quiescence-based artemisinin resistance. ART = artemisinin.

**Figure 3 medicines-09-00002-f003:**
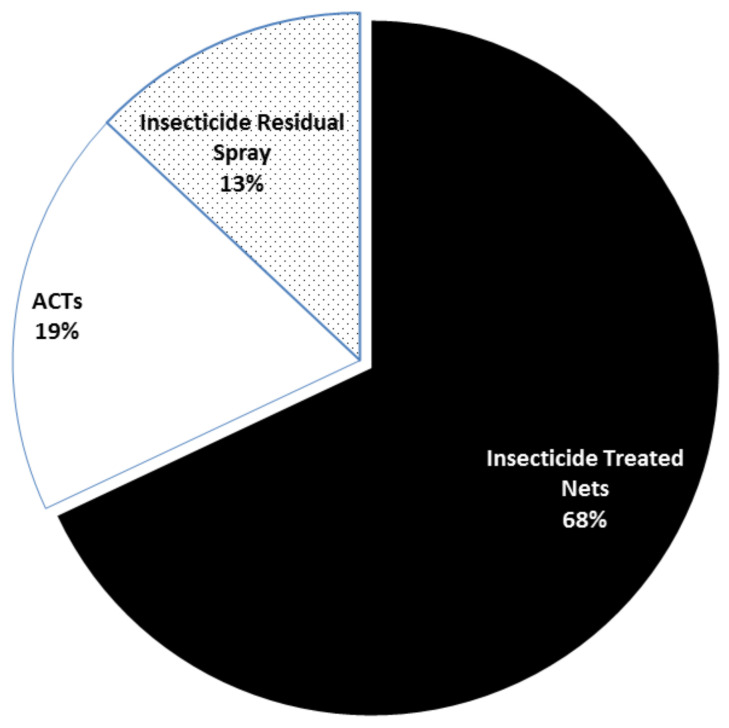
Contribution of different tools to malaria control [69]. As a result of the huge contribution of insecticide treated bed nets, their consistent use is highly advocated in order to reduce or restrict the use of antimalarials in malaria control. This will reduce drug resistance due to drug pressure.

**Table 1 medicines-09-00002-t001:** Antimalarial portfolio.

Antimalarials	Introduction Date	Resistance Date	Genetic Marker	Mechanism of Action	Mechanism of Resistance	References
**Aryl-amino alcohol**						
Quinine	1820	1910	Pfmdr1	Inhibition of heme detoxification	Gene amplification and mutation leading to drug efflux and/or non-binding to target site	[16]
Mefloquine	1977	1982	Pfmdr1			[17]
Lumefantrine	1976		Pfmdr1			[18]
**4-aminoquinolines**						
Chloroquine	1945	1957	Pfcrt, Pfmdr	Inhibition of heme detoxification and redox cycling of heme to and fro the cytosol	Gene mutation: efflux of molecules from food vacuole	[19,20]
Amodiaquine	1948	1990s	Pfcrt, Pfmdr			[18,21]
**Bis-quinoline**						
Piperaquine	1960s	2010	Pfcrt, Pfplasmepsin 2-3 copy number	Inhibition of heme detoxifica-tion and redox cycling of heme to and fro the cytosol	Gene amplification	[22,23]
**Naphthyridine**						
Pyronaridine	1980	-		Inhibition of heme detoxification		[24]
**Antifolates**						
Sulfadoxine	1937	1960s	Pfdhps	Inhibition of folate metabolism and DNA replication	Gene mutation leading to drug binding site modification	[25]
Proguanil/cycloguanil	1948	1949	Pfdhfr			[26]
Pyrimethamine	1952	1960s				
**Endoperoxide**						
Artemisinin and its derivatives (DHA, ATS, ATM)	1972	2008	PfKelch13	Inhibition of heme detoxification and C-C radical formation	Entry into quiescent state	[27]
**Hydroxynaphtoquinones**						
Atovaquone	1996	1996	Pfcytb	Competitive inhibition of Complex iii of the ETC	Modification of binding site on Complex iii/cytochrome b	[26,28]
**Tetracycline antibiotic**						
Doxycycline	1967		SNPs in Pfmdt and PftetQ	Inhibition of protein, nucleotide and deoxynucleotide synthesis	Not yet described	[29,30]

Note: DHA = dihydroartemisinin, ATS = artesunate, ATM = artemether, SNPs = single nucleotide polymorphism.

**Table 2 medicines-09-00002-t002:** Artemisinin-based combinations therapies (ACTs) against *P. falciparum* recommended by the WHO [31,32,33].

ACTs	Region Used	Region of Reported ACT Failure
Artemether-lumefantrine (AL)	Africa, Americas and Middle East	Burkina Faso, Cambodia, Lao People’s Democratic Republic, Thailand and Vietnam
Dihydroartemisinin-piperaquine (DHA-PPQ)	Southeast Asia, China and Africa	Cambodia, Lao People’s Democratic Republic, Thailand and Vietnam
Artesunate-amodiaquine (AS-AQ)	West Africa	Indonesia, Cambodia
Artesunate-mefloquine (AS-MQ)	Southeast Asia and Americas	Cambodia, Lao People’s Democratic Republic, Thailand and Vietnam
Artesunate-sulfadoxine-pyrimethamine (AS-SP)	Southeast Asia, Middle-East and South America	Northeastern India, Somalia and Sudan
Artesunate—pyronaridine (AS-PY)	Southeast Asia	Cambodia, Vietnam

Note: Failure rates of ≥10% were considered according to the WHO therapeutic efficacy studies reports [2].

**Table 3 medicines-09-00002-t003:** Drugs interacting with antimalarials.

Food/Drug	Class	Probable Mechanism of Interaction	Consequences
Indinavir, nelfinavir	Antiretroviral	Inhibits CYP3A4	May increase concentrations of ART and LUM
Imatinib	Anticancer	Inhibits Syk, Lyn, Bcr-Abl	Decrease artemisinin concentration and accelerate ART efficacy
Ritonavir	Antiretroviral	Inhibits CYP2D6 and CYP3A4	May increase concentrations of ART and LUM
Ketoconazole	Antifungal	Inhibits CYP3A4	Shown to cause modest increase in concentration of ART and LUM
Fluconazole	Antifungal	Inhibits CYP3A4	May cause increase in concentration of ART and LUM
Rifampicin, isoniazid	Anti-tuberculosis	Induces CYP3A4	May decrease concentrations of ART and LUM
Nevirapine, efavirenz	Antiretrovirals	Induces CYP3A4	May decrease concentrations of ART and LUM
Phentytoin/phenobarbital /carbamazepine	Anticonvulsants	Induces CYP3A4	May decrease concentrations of ART and LUM

Adapted from [46,47,48]. ART = artemisinin, LUM = lumefantrine, CYP = cytochrome P450.

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
