# Peer review of "Impact of Drug Pressure versus Limited Access to Drug in Malaria Control: The Dilemma"

_medicines, 2022, doi:10.3390/medicines9010002_

Round 1

Reviewer 1 Report

This is an interesting manuscript for a complex topic, the interplay between drug pressure and access to quality assured antimalarial drugs. The authors describe well the complexity of any malaria control programme that has to take into account both issues. I would have liked the authors to discuss more about the role of prompt diagnosis with highly sensitive and specific diagnostic tools. Also, some of the malaria control strategies require the heavy use of antimalarials (chemoprevention strategies), it would be good to discuss about their advantages and disadvantages, and potential alternatives. More recently, triple combinations have been suggested to be one of the solution to ACT resistance, the authors should briefly discuss about this as well. The manuscript has some limitations that need to be addressed before being formally accepted for publication. Below are more specific comments.

1. Introduction

  • Reduced access to drugs is causing deaths as well
  • Other factors that can lead to treatment failure are high baseline parasite density and young age.  
  • Poor access to drugs may be due to not have access to healthcare for populations leaving in remote and/or undeserved areas
  • Poor access could also contribute to resistance. If the drugs are not provided free of charge, patients may not complete the full dose treatment and keep some pills for the next malaria episode, this could also contribute to the emergence and spread of resistance. 

2. Overview of the current antimalarial portfolio

  • Today, many drugs are available for malaria treatment, but only ACTs are recommended to treat uncomplicated falciparum malaria. Chloroquine can still be used for vivax malaria in some regions
  • Please revise table 1, there are some mistakes, I would recommend to review and reference this publication: https://www.sciencedirect.com/science/article/pii/S1473309919302610?via%3Dihub. For example, sulfadoxine is a sulfanomide, not a antifolate
  • Endoperoxides are not targeting the haem detoxification like the aminoquinolines, but rather need haem to be activated. Their mode of action is quite complex as they do not have a single target. it is generally agreed that their primary mode of action is through oxidative stress that will impact proteins and lipids.
  • For artemisinin derivatives, dihydroartemisinin should be included, as this is the active metabolite
  • ACTs are not meant to protect only artemisinin derivatives, but also partner drugs
  • Table 2: So far, there is no confirmed resistance to Artemether lumefantrine and pyronaridine artesunate

3. Limited access to drugs

  • Thee paragraph on cost of drug should also mention that in most malaria endemic countries, antimalarials are also available free of charge through the public health sector
  • Drug interactions: the interaction between vitamin C and antimalarials has never been observed in patients
  • For the paragraph on monotherapies and non ACT combination, it should be mentioned that atovaquone-proguanil is mainly used in travellers, and sulfadoxine-pyrimathamine only for chemoprevention
  • Monotherapies do not reduce exposure, but the parasite is only exposed to one compound making it easier to develop resistance
  • Figure 1: limited access to drugs (use of monotherapies, sub-standard drugs, drug degradation during storage, non-treatment compliance) can also cause resistance

4. Drug pressure

  • Antifolates: pfdhfr mutations are associated with pyrimehtamine resistance and pfdhps mutations with sulfadoxine resistance. The statement on drug with half life is not correct and should not placed in the paragraph for antifolates only. Indeed drug with longer half life will give a wider window for resistance selection when the drug falls under sub-therapeutic levels after treatment. But it does not mean those parasites are more resistant. Also drug with a short half life can develop resistance quite quickly like atovaquone. So, drug half life does matter for resistance development, but also the mode of action.
  • Quinolines:  Pfcrt and Pfmdr1 are only involved in drug efflux, and not interfering with drug binding sites
  • Endoperoxides: PfK13 mutations (validated molecular markers of artemisinin resistance) have been detected in Guyana, Papua New Guinea, India and Uganda. Moreover in Uganda and Rwanda, they have been associated with delayed parasite clearance and higher survival rates in vitro
  • Cross resistance due to drug pressure: Even though not fully understood, artemisinin derivatives are mainly acting by causing oxidative stress, this is not the same mode of action than quinolines that are interfering with haem detoxification. However, artemisinin derivaitve do need haemoglobin degradation to be activated, so this is quite complex, and a direct mechanism of cross resistance cannot be confirmed
  • Proper diagnosis before treatment: Serology is not meant for point of care diagnosis, but rather to assess past exposure to malaria parasites
  • Treatment with only pharmacologically relevant doses: The statement is not correct. Sub optimal doses are the main drivers of resistance, not high doses. The optimal dose is the highest that is well tolerated by patients
  • Recycling of antimalarials: This seems to be a good idea, but one has to take into account the potential rapid re-emergence of resistance when re-introducing a drug, as resistant parasite will still be present at very low levels, but could be rapidly selected by the heavy use of the drug.

Author Response

Please see attachment in the box

Reviewer 2 Report

  1. “ drug interactions, poor and absorption [5]” What is poor and absorption ?

  1. “on the overall malaria treatment failure vis-à-vis it control.” Sentence needs revision.

  1. Is Figure 2 new data presented here or published data or diagrammatic representation ? The way it is presented this remains unclear.
  2. Figure 1 is unclear. Malaria control reduction is not clear to me ? There are also grammatical mistakes
  3. Table 3 needs reference. I am not sure the basis of grapefruit juice inclusion with drugs. Please keep only drugs in this table where evidence is solid.

  1. Table 3 title is not clear
  2. Table 2 : I am not sure the value of mentioning brand owner’s name as separate section.

  1. I am not sure if the evidence really exists how much the access to drug was issue in the overall malaria treatment failure. The role of drug pressure is more clear from laboratory studies. All pathogens develop resistance to drugs at some point.

  1. “Drugs against infectious dis-eases like malaria are among the most falsified drugs in the market especially in developing countries, which fall under malaria endemic regions “ Many drugs are falsified in such markets how it is different from other bacterial infections or other drugs say for COVID19 or cancer.

  1. “Based on this, vitamin C or grapes should be avoided during antimalarial treatment.” Is there any evidence that the dosage due to eating grapes or taking vitamin c tablets has any effect on anti-malarial efficacy. I am not sure if this is studied.

  1. Overall many important reference are missing for example section 4.2.2

Reviewer 3 Report

The review is interesting and it covers many of the aspects related to drug resistance and access to drugs. I would suggest a large revision of the written english 

Moreover, it should be clarified that drug pressure acts on mutations that are randomly acquired by the parasite population and not induced by the drugs. Mutation is different from selection of mutants by the drug pressure and it is also different from spread of drug resistant parasites. These concepts are not clearly expressed.

I also suggest to give a larger overview of all the hypotheses on the mechanisms of artemisinin resistance.

Final comments: major revisions required

Round 2

Reviewer 1 Report

I would like to thank the authors for the revised manuscript, it looks much better now. However I still have few minor comments.

  • Drug interaction is not leading to poor absorption, but rather to increased and decreased drug exposure
  • Atovaquone-proguanil-artesunate will not probably be the first choice in malaria endemic settings for triple therapies. Atovaquone-proguanil (Malarone) is a quite expensive drug that is mainly used for cehmoprophylaxis in travellers. Moreover, resistance to both atovaquone and proguanil can quickly emerge under heavy drug pressure
  • Atovaquone alone is not used in chemoprevention
  • It is not only DHA-PQP that is failing in South East Asia, artesunate mefloquine as well. Moreover, artesunate-amodiaquine and artesunate-sulfadoxine-pyrimethamine are not used due to the high level of resistance to both sulfadoxine-yprimethamine and amodiaquine
  • Table 2: I would also remove the trade names, or add the other trade names that are used for the different drugs
  • Paragraph 4.2.1: please correct transerable, I guess you mean transferable
  • Presumptive treatment is not anymore recommended
  • By dipstick, I guess you mean rapid diagnostic test (RDT)
  • What do you mean by accurate choice of the diagnostic tool ? For patients with fever, RDT and microscopy  are the best tools to diagnose malaria. So there is no choice to make, the available tool should be used, of course with trained and qualified staff

Reviewer 2 Report

The authors answered some of my comments. The title is not so clear. I think the authors want to put access to drugs as a major point. If so, then it needs to be elaborated in detail. The way it is written it is not explained and references are mostly from WHO reports and not original studies. I made this comment before that I am not sure if there is enough proof on drug quality, access to drugs, and drug resistance. The primary reason is how the authors have not discussed in detail. There are broad generalized statements given without discussing specific examples. Further, the language is very poor and correction from a native English speaker is needed before this study can be published.

Reviewer 3 Report

The manuscript has been improved. Still minor spelling errors are present which could be corrected at the proof level.
